# Comment on 'Soil organic stocks are systematically overestimated by misuse of the parameters bulk density and rock fragment content' (Poeplau *et al.*, 2017, SOIL, 3(1), 61-66).

Eleanor Ursula Hobley[1], Brian Murphy[2], Aaron Simmons[3]

[1]Soil Science, Technical University of Munich, Weihenstephan, Germany

[2]NSW Office of Environment and Heritage, Swan Hill, Australia

[3]NSW Dept. Primary Industries, Orange, Australia

*Correspondence to*: Eleanor U. Hobley (Nellie.hobley@wzw.tum.de)

Poeplau et al [2017] recently outlined the systematic overestimation of soil organic carbon (SOC) stocks due to incorrect application of bulk density and rock fragment content in calculation of SOC stocks. Unfortunately, the method they propose to rectify this is associated with a greater error (due to assumption of rock density, extra calculation steps and propagation of errors) than the simpler mass balanced derived equation for SOC stock calculations, outlined below. Using a mass balance approach to C stocks we define:

$$C_{Stock} = Mass\ Proportion_C \cdot \rho \cdot d \qquad (i)$$

Where $C_{stock}$ is the amount of carbon stored in a given soil area (kg m$^{-2}$) and depth, d (cm); Mass Proportion$_C$ is the carbon content of the whole soil (g kg$^{-1}$) and $\rho$ is the bulk density of the whole soil (g cm$^{-3}$).

Using a mass balance approach on the Mass Proportion of C in the whole soil, we obtain:

$$MassProportion_C = C_{Content,\ fine} \cdot Mass\ Proportion_{fine} + C_{Content,\ coarse} \cdot Mass\ Proportion_{coarse} \quad (ii)$$

Where $C_{Content,fine}$ is the mass proportion of C in the fine soil fraction (g kg$^{-1}$), Mass Proportion$_{fine}$ is the mass proportion of the fine soil to the whole soil sample (g kg$^{-1}$) and $C_{Content,\ coarse}$ is the mass proportion of C in the coarse soil fraction (g kg$^{-1}$), Mass Proportion$_{Coarse}$ is the mass proportion of the coarse soil to the whole soil sample (g kg$^{-1}$), generally referred to as the rock content. $C_{Content,\ coarse}$ is assumed to be negligible (i.e. = 0) in all methods, so that the equation (2) simplifies to:

$$MassProportion_C = C_{Content,\ fine} \cdot Mass\ Proportion_{fine} \qquad (iii)$$

The Mass Proportion$_{fine}$ is

$$MassProportion_{fine} = \frac{Mass_{fine}}{Mass_{Total}} = \frac{Mass_{fine}}{Mass_{fine} + Mass_{coarse}} \qquad \text{(iv)}$$

$$= \frac{Mass_{fine} + Mass_{coarse} - Mass_{coarse}}{Mass_{fine} + Mass_{coarse}} \qquad \text{(v)}$$

$$= 1 - Mass\ Proportion_{coarse} \qquad \text{(vi)}$$

Substituting equation (vi) into equation (iii) we obtain:

$$MassProportion_C = C_{Content,\ fine} \cdot (1 - Mass\ Proportion_{coarse}) \qquad \text{(vii)}$$

Substituting equation (vii) into (i) we obtain:

$$C_{Stock} = C_{Content,\ fine} \cdot (1 - Mass\ Proportion_{coarse}) \cdot \rho \cdot d \qquad \text{(viii)}$$

This looks similar to equation (5) in Poeplau *et al.* [2017]. However, they use the volumetric proportion, not the mass proportion of rock fragments, which is mathematically incorrect. They also state that their equation (6) 'resembles' equation (viii). However, their M4 is actually a more convoluted and obtuse equivalent to the commonly known and applied equation (viii) (Ellert and Bettany 1995; Goidts et al. 2009, Mikha et al. 2013; Orgill et al. 2013). This can be shown by combining equations (3) and (6) from Poeplau *et al,* because, as can be shown by combining equations (3) and (6) from Poeplau *et al,* the inclusion of rock density to calculate SOC stocks is unnecessary and redundant.

Equation (viii) is also mathematically equivalent to calculations according to equations (7) and (8) in Poeplau *et al*. However, the recommended use of the mass of fine fraction for the calculations by Poeplau *et al*. also has a greater potential error than using the mass proportion of rocks according to equation (viii). The advantage of using the rock mass to correct the stocks is that rocks are (nearly) entirely conserved during sieving, whereas fine soil mass is lost as dust during sieving, increasing uncertainty in the calculations. In contrast, M4 (equations (3) and (6)) of Poeplau *et al*. requires an estimation of rock density (they recommend assuming a rock density of 2.63 g cm$^{-3}$) to calculate the bulk density of the fine soil sample as well as to adjust for rock content. Rock density depends on parent material, with basalts having higher densities than granites, so that this assumption increases error and uncertainty (Hazelton and Murphy, 2016).

Unfortunately, the additional calculations required also increase the uncertainty of the estimate due to error propagation. Although mathematically equivalent, calculations according to their M4 are therefore less precise due to extra sources of error (derived from either analytical or assumed rock density as well as error propagation). As such, using equation (viii) above, based on the C content of the fine soil, mass proportion of rocks and bulk density in the whole sample will yield the most precise estimate of C stocks.

Unfortunately, the additional calculations required in M4 also increase the uncertainty of the estimate due to error propagation. This can be illustrated by calculating the error terms of both equations. The squared relative error of equation (viii) is:

$$\frac{\sigma_{C_{stock}}^2}{C_{stock}^2} = \frac{\sigma_{C_{content,fine}}^2}{C_{content,fine}^2} + \frac{\sigma_{Mass\ proportion_{Rock}}^2}{Mass\ proportion_{Rock}^2} + \frac{\sigma_{\rho_{Sample}}^2}{\rho_{Sample}^2} + \frac{\sigma_{Depth}^2}{Depth^2}$$

With $Mass\ proportion_{Rock} = \frac{Mass_{Rock}}{Mass_{Sample}}$ and $\rho_{Sample} = \frac{Mass_{Sample}}{Volume_{Sample}}$ we obtain:

$$\frac{\sigma^2_{C_{stock}}}{C_{stock}^2} = \frac{\sigma^2_{C_{content,fine}}}{C_{content,fine}^2} + \frac{\sigma^2_{Mass_{Rock}}}{Mass_{Rock}^2} + 2\frac{\sigma^2_{Mass_{Sample}}}{Mass_{Sample}^2} + \frac{\sigma^2_{Volume_{Sample}}}{Volume_{Sample}^2} + \frac{\sigma^2_{Depth}}{Depth^2}$$

The squared relative error of M4 in Poeplau et al. is:

$$\frac{\sigma^2_{C_{stock}}}{C_{stock}^2} = \frac{\sigma^2_{C_{content,fine}}}{C_{content,fine}^2} + \frac{\sigma^2_{Volume\ proportion_{Rock}}}{Volume\ proportion_{Rock}^2} + \frac{\sigma^2_{\rho_{fine}}}{\rho_{fine}^2} + \frac{\sigma^2_{Depth}}{Depth^2}$$

5  Using the equation 3 in Poeplau et al. for $\rho_{fine}$ and with $Volume\ proportion_{Rock} = \frac{Volume_{Rock}}{Volume_{Sample}}$ we obtain:

$$= \frac{\sigma^2_{C_{content,fine}}}{C_{content,fine}^2} + \frac{\sigma^2_{Volume_{Rock}}}{Volume_{Rock}^2} + \frac{\sigma^2_{Volume_{Sample}}}{Volume_{Sample}^2} + \frac{\sigma^2_{Mass_{Sample}}}{Mass_{Sample}^2} + \frac{\sigma^2_{Volume_{Sample}}}{Volume_{Sample}^2} + 2\frac{\sigma^2_{Mass_{Rock}}}{Mass_{Rock}^2} + \frac{\sigma^2_{\rho_{Rock}}}{\rho_{Rock}^2} + \frac{\sigma^2_{Depth}}{Depth^2}$$

With $\rho_{Rock} = \frac{Mass_{Rock}}{Volume_{Rock}}$ the squared relative error of M4 in Poeplau et al. is therefore:

$$\frac{\sigma^2_{C_{content,fine}}}{C_{content,fine}^2} + 2\frac{\sigma^2_{Volume_{Rock}}}{Volume_{Rock}^2} + 2\frac{\sigma^2_{Volume_{Sample}}}{Volume_{Sample}^2} + \frac{\sigma^2_{Mass_{Sample}}}{Mass_{Sample}^2} + 3\frac{\sigma^2_{Mass_{Rock}}}{Mass_{Rock}^2} + \frac{\sigma^2_{Depth}}{Depth^2}$$

10  As can be seen, M4 has more sources of error than equation (viii). M4 is therefore statistically inferior and should be avoided. This is in line with applying the law of parsimony (Occam's razor) to the problem of SOC stock calculations, which states that when presented with competing answers to a problem, one should choose the one which makes the fewest assumptions. Calculations according to their M4 are therefore less precise due to extra sources of error (derived from either analytical or assumed rock density as well as error propagation). As such, using equation (viii) above, based on the C content of the fine
15  soil, mass proportion of rocks and bulk density in the whole sample will yield the most precise estimate of C stocks.

With regards to eliminating the depth, d, from the calculations (equation (9) in Poeplau *et al.*, suggested by Wendt and Hauser, 2013), it would appear that the error of this method is lower still. However, this is deceptive, because the error associated with sampling a specific depth remains, so that the mathematical simplification does not eliminate the error term.

Of key concern - and not addressed here - is the calculation of SOC stocks in stony soils, as here an accurate estimation of
20  rock content is highly difficult. Estimating rock content from the profile face is highly error prone, because 2D surface areas are not representative of irregular 3D structures, such as rocks. Therefore, estimating rock content from the profile face is not volumetric. Taking larger volumes of sample in very large cores to determine the bulk density of the whole soil would help to alleviate this issue, but would be associated with more field and laboratory work. A systematic study into this issue, similar to the systematic evaluation of sources of error when up-scaling to SOC analyses to landscape stocks (Goidts, van Wesemael &
25  Crucifix, 2009) could help to resolve the issue.

In summary, Poeplau *et al*. have clearly demonstrated the need to adjust for coarse fragments >2 mm in SOC stock calculations. Unfortunately, their recommendation has added some confusion to the correct method of calculation of SOC stocks via the introduction of unfamiliar formulas. Whilst mathematically correct, their formulas are associated with larger errors than the standard equation and are not universally applicable, so present no clear advantage. As such, we recommend the use of equation (viii) for SOC stock calculations.

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
