# Peer review of "Comment on ‘Soil organic stocks are systematically overestimated by misuse of the parameters bulk density and rock fragment content’ (Poeplau *et al.*, 2017, SOIL, 3(1), 61-66)."

_SOIL, 2017_

## Short Comment (SC1) · 22 Feb 2018

1. Generally, the proposed additional equation of Hobley et al. does exactly equal the equations proposed to calculate SOC stocks (7, 8, 9) in Poeplau et al. 2017. It is the mass balance approach (Fine soil stock- FSS - includes bulk density of the whole soil as well as depth of the respective increment) and we argue in the same direction: volume of rocks is not needed to correctly calculate SOC stocks. We are thus questioning if this comment does add clarity to the whole topic or if readers get even more confused because the discussion is turning in circles.

Again:

Equation viii proposed by Hobley et al.:

SOC=C_finesoil×(1-massproportion coarse fraction)×bulkdenity×depth

equals our equations 7 and 8 (Poeplau et al. 2017):

FSS=mass_finesoil/volume_sample ×depth_i

SOCstock=SOCcon_(fine soil)×FSS

Since FSS can also be written as:

FSS=(1-massproportion coarse fraction)×bulkdensity×depth

2. The criticism, that the proposed simplified set of equations (7, 8, 9) would only be possible for single-layer samples but not for subdivided samples is wrong, since FSSi can be calculated for as many layers as needed, of course (FSS0-10, FSS10-20. . .).

3. The argument, that rather coarse soil mass should be measured, while measurement of fine soil is more prone to errors (line 13-17) due to losses of dust is out of focus and beyond the scope of the paper Poeplau et al. 2017: 1. It is not clear, if the error of the coarse fraction is not equally large (e.g. fine soil particles that stick to stones or especially roots). 2. This is a methodological issue of sieving and weighing soil and if anybody has the observation that a significant fraction of fine soil is lost via dust (which is soil specific), fine soil mass can still be easily obtained by total soil mass minus rock fragment mass. The paper Poeplau et al 2017, however, was on the calculations of SOC stocks, i.e. the use of parameters in the equations.

4. It is also explicitly mentioned in our paper (page 62, directly after Eq.6), that if rock fragments fraction is not the volume but the mass fraction, then M3 resembles M4 with equation 6 (by the way, this is IPCC standard). Although the equation proposed by Hobley et al. is thus not explicitly written out, it is mentioned in the text.

5. We disagree, that the volume proportion is mathematically incorrect (as stated in line 9). A proportion is a proportion, may it be mass or volume. If the term 'mathematically incorrect' refers to the whole equation 5, then yes, it is incorrect, and that is what our paper is all about: Showing possible incorrect ways to calculate SOC stocks and helping to avoid them in the future.

6. What remains from the short comment from Hobley et al. is the criticism on proposing 2.6 g cm-3 as a value for rock density. Indeed, this is an approximation of a rock density and this approximation is likely as wrong as the assumption that the rock fraction contains no organic carbon as done by Hobley et al. (line 23) and also in our paper. The message of our paper, just as the message of Hobley et al. is, that rock volume and also rock (and root!) density is actually not needed to calculate SOC stocks when using the mass balance approach (see Poeplau et al. section "Recommended equations to calculate SOC stocks"). However, our approach was to include as many cases (soil sampling methods, data availability. . .) as possible. There might be cases, in which the fine soil is sampled for example with a thin auger, while the rock fragment fraction was estimated on a profile wall. In this case, this estimate is a volumetric one. Then, the bulk density of the fine soil is measured from the auger sample, but the sampled layer has to be reduced by the volume of the rock fragment fraction. This is what M4 does (combining bulk density of the fine soil with a reduced volume of the layer. It is true, that assuming a certain rock fragment density can introduce uncertainty, and we would have been wise to state that as well. What we stated is that e.g. the source of error for neglecting roots is not further discussed, which goes into a similar direction. The equation viii proposed by Hobley et al. is similarly prone to errors: Often it is not clear (e.g. when obtained from archives or literature), what the given bulk density is referring to: bulk density of the fine soil or bulk density of the total soil. In particular in soil with large rock fragments or a large coarse fraction bulk density is often measured only for the fine soil. With the equation set we provided in Poeplau et al. 2017 we insured that any mix up and misuse of bulk density data is not possible.

7. Thus, there are several equations used to calculate SOC stocks that are simply wrong and it was the objective of our paper to point this out. However, there are also several equations that are correct (may it be M3 with mass proportion of the rock fragment instead of volume proportion as written in the text of our paper or in equation viii of Hobley et al., may it be M4, or may it be equations 7, 8 and 9 in our paper). The choice of a specific set of equation depends on sampling strategy and/or data availability. Assuming that stone density in M4 is correctly estimated (or measured!), all approaches come to the same result. Thus, it is not correct that the equation proposed by Hobley et al. "will yield the most precise estimate of C stocks"(p.2 l.23) compared to the other equations.

---

## Referee Comment (RC1) · Anonymous Referee #1 · 14 Mar 2018

I am glad that the discussion on the SOC stock is lively. However, I have to agree with the comment of Poeplau that there is no mathematical difference between the calculations in Poeplau et al and the ones proposed by Hobley et al. Therefore, a further discussion does not necessarily clarify the calculation of SOC stocks. It is vital that authors carefully state the equation for the SOC stock that they use, define the way they express the rock fragment content and last but not least the bulk density. Although this is not explicitly mentioned by Poeplau nor by Hobley, the bulk density of a stony soil is not so easy to estimate. I fear that in many cases the bulk density is estimated from

the weight of a small cylinder inserted in the soil between the rock fragments. Does such an estimate represent the whole soil or fine earth bulk density? In short I would be in favour of ending the discussion, and making sure in our reviews that we carefully consider the way the SOC stock of stony soils is calculated.

––––––––––––––––––––––––––––––

---

## Referee Comment (RC2) · Anonymous Referee #2 · 19 Mar 2018

I agree with both groups of authors that always extra care need to be taken in order to minimize both systematic bias as well as random error in SOC stock calculations and try to get as best as possible a handle on the error propagation effects due to the associated predictions/measurements of bulk density and rock fragment. Although to my understanding (and after reading C. Poeplau's reply), the calculations as proposed in both studies will not yield different results, important elements which may introduce potentially much larger sources of uncertainty are not considered here in this thread and/or do not receive the attention required according to their relative importance, such

as: (i) the fact that often the really large stones are not considered, because they do not fit within the sample (see also comment R1 –> the rock fragment content in the soil samples are often not representative for the rock fragment content on the site), (ii) the effects of using equal mass versus equal depth basis for SOC stock predictions, (iii) the effect of using PTFs, (iv) the SOC content determination method (e.g. W&B, LOI, dry combustion), (v) the sampling strategy, (vi) the interpolation technique / modelling or statistical approach considered in the context of landscape level stock estimations, ...
So, when taking all these other sources of uncertainty into consideration, one may state that the relevance/importance of the discussion as presented in this thread reduces considerably. Consequently, despite the fact that it is certainly worth to have a debate on this matter, I have my serious doubts if the interactive comment interface of a high impact factor journal, such as SOIL, is the most appropriate environment to hold this debate, and therefore, I suggest to close the discussion here (and that the two groups of authors potentially get in touch and write a common review paper on the various sources of factors affecting the uncertainty of SOC stock predictions).

---

## Short Comment (SC2) · 26 Mar 2018

As pointed out by all reviewers, it is nice to see that a lively debate about the correct calculation of SOC stock is underway. The reviewers and authors have pointed out some valid points, but despite the suggestion that the debate be closed (Referees 1 and 2), there remain a few points of clarification.

Firstly we wish to note that we did not state that the volumetric proportion is incorrect (Poeplau response), but that it's use is incorrect. This statement is correct.

[Figure]

Secondly, when we approached the authors in June 2017 to suggest we prepare a joint publication on this, as suggested by Referee 2, they stated that they did not think it was necessary.

Chiefly, Poeplau et al. state in their paper that M4, which they use as the 'correct' standard for comparing, is 'the closest approximation to reality'. They also state in their response that this is the same as the IPCC method (why then, did they not use the IPCC method as their gold standard for comparison?). M4 is however, not entirely equivalent to the equation (viii) derived in our comment, which is illustrated by comparing the errors of the two methods (see pdf supplement for the error equations and their derivation).

The equation M4 of Poeplau et al. has more sources of error than the equation (viii) in our comment and M4 should therefore be avoided. This is the in line with applying the law of parsimony (Occam's razor) to the problem of SOC stock calculations, which states that when presented with competing answers to a problem, one should choose the one which makes the fewest assumptions. As pointed out by the reviewers, there are numerous sources of error in SOC stock calculations, including one's which have not been considered here, and we as scientists must always aim to minimize or eliminate them.

On this note, equation 9 of Poeplau et al. has the least theoretical error, although the practical error associated with sampling to a specific depth remains, so that's is simplicity is partially deceptive. The authors are indeed correct in their assertion that this can be used for multiple depth samples. Equation 9 from Poeplau et al. is therefore an attractive option, though care must be taken that the calculation of the fine soil stock is not derived from the equations presented in their M4 due to the high associated errors of this method.

Of key concern - and not addressed here - is the calculation of SOC stocks in stony soils, as here an accurate estimation of rock content is highly difficult. Estimating rock

content from the profile face is highly error prone, because 2D surface areas are not representative of irregular 3D structures, such as rocks. Therefore, estimating rock content from the profile face is not volumetric, as stated by Poeplau. Taking larger volumes of sample in larger cores to determine the bulk density of the whole soil would help to alleviate this issue, but would be associated with more field and laboratory work. A systematic study into this issue, similar to the systematic evaluation of sources of error when up-scaling to SOC analyses to landscape stocks (Goidts, van Wesemael & Crucifix, Europ. J-. Soil Science, 2009, doi: 10.1111/j.1365-2389.2009.01157.x) could help to resolve the issue.

Please also note the supplement to this comment:
https://www.soil-discuss.net/soil-2017-23/soil-2017-23-SC2-supplement.pdf

**Supplement:**

The error of equation (viii) derived in our comment is:

$$\frac{\sigma^2_{C_{stock}}}{C_{stock}{}^2} = \frac{\sigma^2_{C_{content,fine}}}{C_{content,fine}{}^2} + \frac{\sigma^2_{Mass\ proportion_{Rock}}}{Mass\ proportion_{Rock}{}^2} + \frac{\sigma^2_{\rho_{Sample}}}{\rho_{Sample}{}^2} + \frac{\sigma^2_{Depth}}{Depth^2}$$

With $Mass\ proportion_{Rock} = \frac{Mass_{Rock}}{Mass_{Sample}}$ and $\rho_{Sample} = \frac{Mass_{Sample}}{Volume_{Sample}}$ we obtain:

$$= \frac{\sigma^2_{C_{content,fine}}}{C_{content,fine}{}^2} + \frac{\sigma^2_{Mass_{Rock}}}{Mass_{Rock}{}^2} + 2\frac{\sigma^2_{Mass_{Sample}}}{Mass_{Sample}{}^2} + \frac{\sigma^2_{Volume_{Sample}}}{Volume_{Sample}{}^2} + \frac{\sigma^2_{Depth}}{Depth^2}$$

The error of M4 in Poeplau et al. is:

$$\frac{\sigma^2_{C_{stock}}}{C_{stock}{}^2} = \frac{\sigma^2_{C_{content,fine}}}{C_{content,fine}{}^2} + \frac{\sigma^2_{Volume\ proportion_{Rock}}}{Volume\ proportion_{Rock}{}^2} + \frac{\sigma^2_{\rho_{fine}}}{\rho_{fine}{}^2} + \frac{\sigma^2_{Depth}}{Depth^2}$$

Using the equation 3 in Poeplau et al. for $\rho_{fine}$ and with $Volume\ proportion_{Rock} = \frac{Volume_{Rock}}{Volume_{Sample}}$

we obtain:

$$= \frac{\sigma^2_{C_{content,fine}}}{C_{content,fine}{}^2} + \frac{\sigma^2_{Volume_{Rock}}}{Volume_{Rock}{}^2} + \frac{\sigma^2_{Volume_{Sample}}}{Volume_{Sample}{}^2} + \frac{\sigma^2_{Mass_{Sample}}}{Mass_{Sample}{}^2} + \frac{\sigma^2_{Volume_{Sample}}}{Volume_{Sample}{}^2}$$

$$+ 2\frac{\sigma^2_{Mass_{Rock}}}{Mass_{Rock}{}^2} + \frac{\sigma^2_{\rho_{Rock}}}{\rho_{Rock}{}^2} + \frac{\sigma^2_{Depth}}{Depth^2}$$

With $\rho_{Rock} = \frac{Mass_{Rock}}{Volume_{Rock}}$ the error of M4 in Poeplau et al. is therefore:

$$= \frac{\sigma^2_{C_{content,fine}}}{C_{content,fine}{}^2} + 2\frac{\sigma^2_{Volume_{Rock}}}{Volume_{Rock}{}^2} + 2\frac{\sigma^2_{Volume_{Sample}}}{Volume_{Sample}{}^2} + \frac{\sigma^2_{Mass_{Sample}}}{Mass_{Sample}{}^2} + 3\frac{\sigma^2_{Mass_{Rock}}}{Mass_{Rock}{}^2} + \frac{\sigma^2_{Depth}}{Depth^2}$$

Therefore, there are extra and more numerous errors associated with the Poeplau M4 than with equation (viii).

---

## Author Comment (AC1) · 7 May 2018

As pointed out by all reviewers, it is nice to see that a lively debate about the correct calculation of SOC stock is underway. The reviewers and authors have pointed out some valid points, but despite the suggestion that the debate be closed (Referees 1 and 2), there remain a few points of clarification.

Firstly we wish to note that we did not state that the volumetric proportion is incorrect (Poeplau response), but that it's use is incorrect.

[Figure]

Secondly, when we approached the authors in June 2017 to suggest we prepare a publication on their paper, as suggested by Referee 2, they stated that they did not think it was necessary. Chiefly, Poeplau et al. state in their paper that M4, which they use as the 'correct' standard for comparing, is 'the closest approximation to reality'. They also state in their response that M4 with equation 6 is the same as the IPCC standard. Although we are not sure which specific equation is referred to, we ask ourselves why then did they not use it as their gold standard for comparison? Unfortunately, we were unable to easily identify the referred to IPCC standard. The IPCC equation to calculate the change in SOC stocks available at https://www.ipcc-nggip.iges.or.jp/public/2006gl/pdf/4_Volume4/V4_14_An2_SumEqua.pdf, but it does not deal with initial stock calculations, bulk density or rock fragments. The ISO standard equation for calculating SOC stocks is shown in Bispo et al. (2017) and Cotching et al. (2013), and this was the standard used for the Australian Soil Carbon Research Program (SCaRP, Baldock et al. 2013). It can be shown that M4 and that used in the SCaRP program yield mathematically equivalent results. M4 is however, not statistically equivalent to the derived (IPCC) equation (viii) in our comment, which is illustrated by comparing the errors of the two methods, shown in the attachment and full response.

The equation M4 of Poeplau et al. therefore has more sources of error than equation (viii) in our comment. M4 is therefore statistically inferior and should be avoided. This is in line with applying the law of parsimony (Occam's razor) to the problem of SOC stock calculations, which states that when presented with competing answers to a problem, one should choose the one which makes the fewest assumptions. This is important, because as author's we have received reviewers comments criticising the use of the statistically superior equation (viii) whilst advocating the use of the statistically inferior M4. We hope that this proof now serves as an adequate rebuttal of any such future comments once and for all. As pointed out by the current reviewers, there are numerous sources of error in SOC stock calculations, including one's which have not been considered here, and we as scientists must always aim to minimize or eliminate them.

A quick conversion of the FSS values and Equations 7 and 8 in Poeplau et al. can be readily shown to be equivalent to the standard equation in Bispo et al. (2017) and Cotching et al. (2013), so one wonders if it is such a revelation. While equation 9 of Poeplau et al., which was also introduced in equation 4 of Wendt and Hauser (2013), has the least theoretical error, the practical error associated with sampling to a specific depth remains, so that's is simplicity is partially deceptive. The authors are indeed correct in their assertion that this can be used for multiple depth samples, but they have not clarified the units to be used and this has left some confusion. Presumably the numerator is intended to be measured in g? While Equation 9 from Poeplau et al. is a seemingly attractive option, care must be taken that the calculation of the fine soil stock is not derived from the equations presented in their M4 due to the statistical inferiority of this method.

Of key concern - and not addressed here - is the calculation of SOC stocks in stony soils, as here an accurate estimation of rock content is highly difficult. Estimating rock content from the profile face is highly error prone, because 2D surface areas are not representative of irregular 3D structures, such as rocks. Therefore, estimating rock content from the profile face is not volumetric. Taking larger volumes of sample in very large cores to determine the bulk density of the whole soil would help to alleviate this issue, but would be associated with more field and laboratory work. A systematic study into this issue, similar to the systematic evaluation of sources of error when up-scaling to SOC analyses to landscape stocks (Goidts, van Wesemael & Crucifix, Europ. J-. Soil Science, 2009, doi: 10.1111/j.1365-2389.2009.01157.x) could help to resolve the issue.

References: Baldock. J, McDonald, L and Sanderman, J. (2013). Forward: Special Issue : Soil carbon in Australia's Agricultural lands. Soil Research 51, (i). Bispo, A et al. (2017). Accounting for carbon stocks in soils and measuring GHGs emission fluxes from soils: Do we have necessary standards? Frontiers in Environmental Science 5, 41 Cotching, WE, Oliver, G, Downie, M, Corkrey, R and Doyle, RB. (2013). Land use

and management influences on surface organic carbon in Tasmania. Soil Research 51, 615 – 630. Wendt JW, Hauser S (2013) An equivalent soil mass procedure for monitoring soil organic carbon in multiple soil layers. European Journal of Soil Science 64(1): 58-65

Please also note the supplement to this comment:
https://www.soil-discuss.net/soil-2017-23/soil-2017-23-AC1-supplement.pdf

———————————————————————

---

## Author Comment (AC2) · 7 May 2018

Response to all comments

As pointed out by all reviewers, it is nice to see that a lively debate about the correct calculation of SOC stock is underway. The reviewers and authors have pointed out some valid points, but despite the suggestion that the debate be closed (Referees 1 and 2), there remain a few points of clarification.

Firstly we wish to note that we did not state that the volumetric proportion is incorrect (Poeplau response), but that it's use is incorrect.

Secondly, when we approached the authors in June 2017 to suggest we prepare a publication on their paper, as suggested by Referee 2, they stated that they did not think it was necessary.

Chiefly, Poeplau et al. state in their paper that M4, which they use as the 'correct' standard for comparing, is 'the closest approximation to reality'. They also state in their response that M4 with equation 6 is the same as the IPCC standard. Although we are not sure which specific equation is referred to, we ask ourselves why then did they not use it as their gold standard for comparison?

Unfortunately, we were unable to easily identify the referred to IPCC standard. The IPCC equation to calculate the change in SOC stocks available at https://www.ipcc-nggip.iges.or.jp/public/2006gl/pdf/4_Volume4/V4_14_An2_SumEqua.pdf, but it does not deal with initial stock calculations, bulk density or rock fragments. The ISO standard equation for calculating SOC stocks is shown in Bispo et al. (2017) and Cotching et al. (2013), and this was the standard used for the Australian Soil Carbon Research Program (SCaRP, Baldock et al. 2013). It can be shown that M4 and that used in the SCaRP program yield mathematically equivalent results. M4 is however, not statistically equivalent to the derived (IPCC) equation (viii) in our comment, which is illustrated by comparing the errors of the two methods:

The squared relative error of equation (viii) derived in our comment is:

$$\frac{\sigma^2_{C_{stock}}}{C_{stock}^2} = \frac{\sigma^2_{C_{content,fine}}}{C_{content,fine}^2} + \frac{\sigma^2_{Mass\ proportion_{Rock}}}{Mass\ proportion_{Rock}^2} + \frac{\sigma^2_{\rho_{Sample}}}{\rho_{Sample}^2} + \frac{\sigma^2_{Depth}}{Depth^2}$$

With $Mass\ proportion_{Rock} = \frac{Mass_{Rock}}{Mass_{Sample}}$ and $\rho_{Sample} = \frac{Mass_{Sample}}{Volume_{Sample}}$ we obtain:

$$\frac{\sigma^2_{C_{stock}}}{C_{stock}^2} = \frac{\sigma^2_{C_{content,fine}}}{C_{content,fine}^2} + \frac{\sigma^2_{Mass_{Rock}}}{Mass_{Rock}^2} + 2\frac{\sigma^2_{Mass_{Sample}}}{Mass_{Sample}^2} + \frac{\sigma^2_{Volume_{Sample}}}{Volume_{Sample}^2} + \frac{\sigma^2_{Depth}}{Depth^2}$$

The error of M4 in Poeplau et al. is:

$$\frac{\sigma^2_{C_{stock}}}{C_{stock}^2} = \frac{\sigma^2_{C_{content,fine}}}{C_{content,fine}^2} + \frac{\sigma^2_{Volume\ proportion_{Rock}}}{Volume\ proportion_{Rock}^2} + \frac{\sigma^2_{\rho_{fine}}}{\rho_{fine}^2} + \frac{\sigma^2_{Depth}}{Depth^2}$$

Using the equation 3 in Poeplau et al. for $\rho_{fine}$ and with $Volume\ proportion_{Rock} = \frac{Volume_{Rock}}{Volume_{Sample}}$

we obtain:

$$= \frac{\sigma^2_{C_{content,fine}}}{C_{content,fine}^2} + \frac{\sigma^2_{Volume_{Rock}}}{Volume_{Rock}^2} + \frac{\sigma^2_{Volume_{Sample}}}{Volume_{Sample}^2} + \frac{\sigma^2_{Mass_{Sample}}}{Mass_{Sample}^2} + \frac{\sigma^2_{Volume_{Sample}}}{Volume_{Sample}^2}$$
$$+ 2\frac{\sigma^2_{Mass_{Rock}}}{Mass_{Rock}^2} + \frac{\sigma^2_{\rho_{Rock}}}{\rho_{Rock}^2} + \frac{\sigma^2_{Depth}}{Depth^2}$$

With $\rho_{Rock} = \frac{Mass_{Rock}}{Volume_{Rock}}$ the squared relative error of M4 in Poeplau et al. is therefore:

$$\frac{\sigma^2_{C_{content,fine}}}{C_{content,fine}^2} + 2\frac{\sigma^2_{Volume_{Rock}}}{Volume_{Rock}^2} + 2\frac{\sigma^2_{Volume_{Sample}}}{Volume_{Sample}^2} + \frac{\sigma^2_{Mass_{Sample}}}{Mass_{Sample}^2} + 3\frac{\sigma^2_{Mass_{Rock}}}{Mass_{Rock}^2} + \frac{\sigma^2_{Depth}}{Depth^2}$$

The equation M4 of Poeplau et al. therefore has more sources of error than equation (viii) in our

comment.  M4 is therefore statistically inferior and should be avoided. This is in line with applying

the law of parsimony (Occam's razor) to the problem of SOC stock calculations, which states that

when presented with competing answers to a problem, one should choose the one which makes the

fewest assumptions. This is important, because as author's we have received reviewers comments

criticising the use of the statistically superior equation (viii) whilst advocating the use of the

statistically inferior M4. We hope that this proof now serves as an adequate rebuttal of any such

future comments once and for all. As pointed out by the current reviewers, there are numerous sources of error in SOC stock calculations, including one's which have not been considered here, and we as scientists must always aim to minimize or eliminate them.

A quick conversion of the FSS values and Equations 7 and 8 in Poeplau et al. can be readily shown to be equivalent to the standard equation in Bispo et al. (2017) and Cotching et al. (2013), so one wonders if it is such a revelation. While equation 9 of Poeplau et al., which was also introduced in equation 4 of Wendt and Hauser (2013), has the least theoretical error, the practical error associated with sampling to a specific depth remains, so that's is simplicity is partially deceptive. The authors are indeed correct in their assertion that this can be used for multiple depth samples, but they have not clarified the units to be used and this has left some confusion. Presumably the numerator is intended to be measured in g? While Equation 9 from Poeplau et al. is a seemingly attractive option, care must be taken that the calculation of the fine soil stock is not derived from the equations presented in their M4 due to the statistical inferiority of this method.

Of key concern - and not addressed here - is the calculation of SOC stocks in stony soils, as here an accurate estimation of rock content is highly difficult. Estimating rock content from the profile face is highly error prone, because 2D surface areas are not representative of irregular 3D structures, such as rocks. Therefore, estimating rock content from the profile face is not volumetric. Taking larger volumes of sample in very large cores to determine the bulk density of the whole soil would help to alleviate this issue, but would be associated with more field and laboratory work. A systematic study into this issue, similar to the systematic evaluation of sources of error when up-scaling to SOC analyses to landscape stocks (Goidts, van Wesemael & Crucifix, Europ. J-. Soil Science, 2009, doi: 10.1111/j.1365-2389.2009.01157.x) could help to resolve the issue.

References:
Baldock. J, McDonald, L and Sanderman, J. (2013). Forward: Special Issue : Soil carbon in Australia's Agricultural lands. Soil Research 51, (i).
Bispo, A et al. (2017). Accounting for carbon stocks in soils and measuring GHGs emission fluxes from soils: Do we have necessary standards? Frontiers in Environmental Science 5, 41

Cotching, WE, Oliver, G, Downie, M, Corkrey, R and Doyle, RB. (2013).  Land use and management influences on surface organic carbon in Tasmania.  Soil Research 51, 615 – 630.

Wendt JW, Hauser S (2013) An equivalent soil mass procedure for monitoring soil organic carbon in multiple soil layers. European Journal of Soil Science 64(1): 58-65

---

## Author Comment (AC3) · 7 May 2018

Thank you for taking the time to review our comments. We have responded in full in a supplement, due to difficulties uploading equations into the Copernicus comments systems.

Many thanks Hobley, Wilson and Simmons

Please also note the supplement to this comment:

[Figure]

https://www.soil-discuss.net/soil-2017-23/soil-2017-23-AC3-supplement.pdf

---

## Author Response (AR1)

**Response to reviewers and editors comments**

As pointed out by all reviewers, it is nice to see that a lively debate about the correct calculation of SOC stock is underway. The reviewers and authors have pointed out some valid points, but despite the suggestion that the debate be closed (Referees 1 and 2), there remain a few points of clarification.

5    Chiefly, Poeplau et al. state in their paper that M4, which they use as the 'correct' standard for comparing, is 'the closest approximation to reality'. They also state in their response that M4 with equation 6 is the same as the IPCC standard. M4 is however, not statistically equivalent to the derived equation (viii) in our comment, which is illustrated by a comparison of the errors of their M4 compared with the derived equation 8 in our comment. This information has been added to P2 L25ff to demonstrate this point.

10   The authors are indeed correct in their assertion that this can be used for multiple depth samples, but they have not clarified the units to be used and this has left some confusion. Nevertheless, this was deleted from the manuscript (P2 L24) for the sake of brevity.

With regards the estimation of rock content from the profile, we have added a section to discuss rock content estimation P3 L17-23).

15   Lastly, further literature has been added to support the use of equation 8 for calculations of SOC stocks.

[revised manuscript text omitted]